# Indicators of HSV1 Infection, ECM–Receptor Interaction, and Chromatin Modulation in a Nuclear Family with Schizophrenia

**DOI:** 10.3390/jpm13091392

**Published:** 2023-09-18

**Authors:** Yen-Chen Huang, Lieh-Yung Ping, Shih-Hsin Hsu, Hsin-Yao Tsai, Min-Chih Cheng

**Affiliations:** Department of Psychiatry, Yuli Branch, Taipei Veterans General Hospital, Hualien 98142, Taiwan; ianh1981@gmail.com (Y.-C.H.); minipyng@gmail.com (L.-Y.P.); filvhsu@gmail.com (S.-H.H.); ashleytsai0808@gmail.com (H.-Y.T.)

**Keywords:** schizophrenia, family, whole-exome sequencing, rare mutation, pathway analysis

## Abstract

Schizophrenia (SCZ) is a complex psychiatric disorder with high heritability; identifying risk genes is essential for deciphering the disorder’s pathogenesis and developing novel treatments. Using whole-exome sequencing, we screened for mutations within protein-coding sequences in a single family of patients with SCZ. In a pathway enrichment analysis, we found multiple transmitted variant genes associated with two KEGG pathways: herpes simplex virus 1 (HSV1) infection and the extracellular matrix (ECM)–receptor interaction. When searching for rare variants, six variants, *SLC6A19*^p.L541R^, *CYP2E1*^p.T376S^, *NAT10*^p.E811D^, *N4BP1*^p.L7V^, *CBX2*^p.S520C^, and *ZNF460*^p.K190E^, segregated with SCZ. A bioinformatic analysis showed that three of these mutated genes were associated with chromatin modulation. We found that HSV1 infection, ECM–receptor interaction pathways, and epigenetic mechanisms may contribute to the pathogenesis of SCZ in certain families. The identified polygenetic risk factors from the sample family provide distinctive underlying biological mechanisms of the pathophysiology of SCZ and may be useful in clinical practice and patient care.

## 1. Introduction

Schizophrenia (SCZ) is a chronic debilitating mental disorder characterized by abnormal perception, thought disturbances, bizarre behavior, and cognitive deficits [1]. This disorder affects approximately 1% of the general population worldwide. Genetic factors play a critical role in its etiology; the heritability of SCZ has been estimated at 70–80%, with numerous transmitted variants possibly involved [2]. However, discovering genes directly responsible for SCZ is challenging because of its genetic complexity and the epigenetic mechanisms involved, which possibly act as downstream effectors of environmental signals [3].

The current hypotheses of the genetic basis for SCZ include rare deleterious mutations with large clinical penetrance in some patients, psychosis in families, or the addition of multiple common polymorphisms with minor effects [4,5]. Identifying common and rare variants associated with SCZ could increase our understanding of the neurobiology of psychiatric disorders. In this regard, a genome-wide association (GWAS) study identified 108 SCZ-associated genetic loci [4]. Additionally, multiple rare or ultrarare variants, such as loss-of-function, missense, and chromosomal abnormalities, have been associated with SCZ [5,6]. These studies indicate that the genetic underpinnings of SCZ are very complex and heterogeneous. One strategy to better understand the etiology of this disorder is through a family-based analysis of the contribution of rare and common genetic variants to the disorder with whole-exome sequencing (WES) [7,8,9].

Here, we discovered variants in protein-coding sequences in a schizophrenic family with WES, which we confirmed with Sanger sequencing. The identified variants in the familial form of SCZ could provide a new landscape of the genetic architecture and alternative treatment targets for SCZ.

## 2. Materials and Methods

### 2.1. Sample Family Recruitment

The sample family was a nuclear family, including an unaffected father and a mother, son, and daughter with SCZ. All subjects were Han Chinese from Taiwan. The unaffected father and the two children were recruited for the WES experiment. All affected family members were diagnosed according to the Diagnostic and Statistical Manual of Mental Disorders, Fifth Edition (DSM-V) criteria. The study was approved by the ethics committee of the Antai Tian-Sheng Memorial Hospital Institutional Review Board; written informed consent was obtained after all procedures were fully explained to the participants.

### 2.2. WES

Genomic DNA was prepared from white blood cells (WBCs) using a Gentra Puregene Blood Kit (QIAGEN). The quality of the genomic DNA was assessed on a NanoDrop One spectrophotometer (Thermo Fisher Scientific, Waltham, MA, USA). The 260/280 ratio for genomic DNA was ~1.8. The DNA library was prepared using 1 μg of genomic DNA from each sample using an xGenTM Exome Research Panel V2 (IDT, San Diego, CA, USA) following the manufacturer’s protocols with added index codes to attribute sequences to each sample. The library quality was assessed on a Qubit 4.0 fluorometer and Agilent bioanalyzer system. Sequencing was conducted on an Illumina NovaSeq 6000 platform to generate paired 150 bp reads, and the read coverage depth was 22×. The paired-end reads were analyzed for mapping to the human reference genome using the Burrows–Wheeler Aligner (v2.2.1) with default settings. The aligned reads were subsequently processed with a Genome Analysis ToolKit (GATK) (v4.2.1.0), including MarkDuplicates and Base Quality Score Recalibration. Variant calling for single-nucleotide variants (SNVs) and small insertion/deletion mutations was performed using a HaplotypeCaller. High-confidence variants were then obtained with the standard GATK Variant Quality Score Recalibration tool and annotated with ANNOVAR (v2020), SnpEff (v4.3), and VEP (v100.4); the data were then integrated using an in-house algorithm. Copy-number variations and structural variations were called using Control-FREEC (v11.6) and Manta (v1.6.0); next, the variants were annotated with AnnotSV (v3.0.9).

### 2.3. Mutation Validation with PCR-Based Sequencing

All PCR primer sequences were designed using the Primer3 website (http://bioinfo.ut.ee/primer3-0.4.0/primer3/, accessed on 10 August 2022), and the primer sequences, optimal annealing temperatures, and size of each amplicon are available on request. In a standard PCR reaction, genomic DNA (50 ng) was amplified in a reaction volume of 25 μL containing 2× PCR buffer for KOD FX, 2 mM dNTPs, and 1.0 U/μL of KOD FX (TOYOBO Co., Ltd., Osaka-Shi, Japan). The PCR cycling condition consisted of an initial denaturation at 95 °C for 5 min, followed by 30 cycles of 95 °C for 1 min, and an optimal annealing temperature of each target for 1 min and 72 °C for 1 min. For sequencing, aliquots of PCR products were purified using an Illustra™ ExoProStar™ 1-Step Kit (GE Healthcare Bio-Sciences, Piscataway, NJ, USA) and sequenced using a BigDye™ Terminator v3.1 Cycle Sequencing Kit on a SeqStudio DNA Analyzer (Thermo Fisher Scientific) according to the manufacturer’s protocols. The identified mutations were verified with repeated PCR and sequencing in both directions.

### 2.4. Bioinformatic Analysis

The variant coding effect was checked using publicly available prediction programs: Polyphen-2 (http://genetics.bwh.harvard.edu/pph2/, accessed on 8 August 2022), SIFT (http://sift.bii.a-star.edu.sg/, accessed on 8 August 2022), and CADD (https://cadd.gs.washington.edu/, accessed on 8 August 2022). We also determined whether the mutations were documented in the public databases gnomAD (https://gnomad.broadinstitute.org/, accessed on 8 August 2022) and Taiwan BioBank (https://taiwanview.twbiobank.org.tw, accessed on 8 August 2022). For the pathway analysis, genes with identified variants were uploaded to the database for annotation, visualization, and integrated discovery (DAVID v6.8, https://david.ncifcrf.gov/tools.jsp, accessed on 1 August 2022) using the official gene symbol method. Genes with identified variants associated with the Kyoto Encyclopedia of Genes and Genomes (KEGG) pathways were highlighted and annotated [10].

## 3. Results

### 3.1. Clinical Manifestations of SCZ in the Study Family

Figure 1 shows the nuclear family with an unaffected father (I:1), a mother with SCZ (I:2), a son with SCZ (II:1), and a daughter with SCZ (II:2). The SCZ seemed to be transmitted following a dominant inheritance pattern. At the time of the study, the father (I-1) was 61 years old and had no history of psychiatric disorders. The mother (I-2) died in an accident ten years ago. The elder son (II-1) was 30 years old and was born full-term without remarkable events. He began experiencing psychotic symptoms at the age of 14 years. His psychotic symptoms included auditory hallucinations, bizarre behavior, and aggressive behavior toward his family. He had no history of illicit drug abuse, head injury, or seizure attack. He had poor insight and poor drug adherence. As a result, he was hospitalized in psychiatric wards several times due to the relapse of his symptoms. He led a lazy lifestyle with poor socio-occupational functioning. Finally, he was admitted to a psychiatric day ward for long-term rehabilitation. The younger sister (II-2) was 29 years old and manifested auditory hallucinations, bizarre delusions (about aliens and gods), bizarre behaviors, persecutory delusion, and psychomotor agitation from the age of 20 years when she was diagnosed with SCZ. She had marked positive symptoms with multiple relapses, partly due to poor compliance with medication. Later, she did not respond very well to antipsychotic drugs; therefore, clozapine was prescribed in combination with other antipsychotics. She was finally hospitalized in a chronic psychiatric ward due to unremitted psychotic symptoms and poor family support. The clinical findings for the patients are shown in Appendix A.

### 3.2. WES Analysis

A WES analysis was conducted on peripheral WBC samples from the three family members (I:1, II:1, and II:2), and the reads were mapped against the hg38 human reference genome. The statistics of the sequencing reads and the number of identified variants after variant calling for SNVs are listed in Appendix A. A list of complete variants in the study samples is available on request. Next, the identified variants were further filtered to exclude non-protein-modifying variants.

### 3.3. Autosomal Dominant Analysis and Rare Mutation Identification

An autosomal dominant transmission analysis identified 951 protein-altering variants (within 736 genes) in the two affected children (II:1 and II:2) absent in the unaffected father (I:1) (Appendix A). Furthermore, 736 genes were uploaded to the DAVID database for pathway analysis. According to the DAVID analysis (after multiple tests), multiple genes were associated with the following two KEGG pathways: herpes simplex virus 1 (HSV1) infection (hsa05168) and the extracellular matrix (ECM)–receptor interaction (hsa04512) (Table 1). The detailed genetic and bioinformatic information of the variants identified in these two KEGG pathways is listed in Table 1.

Among the 951 protein-altering variants segregated with SCZ, 55 had MAFs < 0.5% in the gnomAD and Taiwan BioBank databases, of which six variants (*SLC6A19*^p.L541R^, *CYP2E1*^p.T376S^, *NAT10*^p.E811D^, *N4BP1*^p.L7V^, *CBX2*^p.S520C^, and *ZNF460*^p.K190E^) were not reported in either of the two databases (Table 2). Sanger sequencing confirmed the presence of these six rarely transmitted protein-altering variants in the sample family (Figure 2).

## 4. Discussion

Identifying the specific genetic susceptibility and polygenetic risk factors for a family with SCZ is essential for establishing the molecular diagnosis, providing insight into the pathogenesis, and guiding the personalized treatment for each affected patient [11]. In this case, to search for the genetic underpinnings, we conducted a WES analysis of a nuclear family with an unaffected father, a son with SCZ, and a daughter with SCZ. Because the mother with SCZ died in an accident ten years ago, we did not collect her sample for the WES analysis. Given that this sample family is a classical SCZ family with a dominant inheritance pattern, we conducted an autosomal dominant transmission analysis to identify protein-altering variants in the two affected children but absent in the unaffected father. After the autosomal dominant analysis, we found multiple transmitted variant genes associated with two KEGG pathways: HSV1 infection and the ECM–receptor interaction. In addition, we detected six variants (*SLC6A19*^p.L541R^, *CYP2E1*^p.T376S^, *NAT10*^p.E811D^, *N4BP1*^p.L7V^, *CBX2*^p.S520C^, and *ZNF460*^p.K190E^) absent in the gnomAD and Taiwan BioBank databases, which indicated that these coding variants may be ultrarare and confer a risk for SCZ in this sample family. Among these ultrarare variants, three (*CYP2E1*^p.T376S^, *NAT10*^p.E811D^, and *CBX2*^p.S520C^) may be associated with epigenetic mechanisms in the pathogenesis of SCZ. The above polygenetic risk factors we identified from the sample family provide distinctive underlying biological mechanisms for the pathophysiology of SCZ and may be helpful in clinical practice and patient care.

Accumulating evidence from genetic and epidemiologic studies demonstrates that the link between immunological processes and neuropsychiatric disorders has increased [12,13,14]. For example, an inflammatory state in the brain plays a role in developing neuropsychiatric disorders [12]. A meta-analysis observed a positive association between non-neurological autoimmune disorders and psychosis [15]. Furthermore, previous research highlighted a potential role for the immune system in autism spectrum disorder [14]. Thus, it seems there is a role of immune systems in neuropsychiatric disorders, and understanding how they interact can improve our understanding of neuropsychiatric disorders and give rise to alternative treatments in psychiatry. Serological evidence has demonstrated that exposure to the neurotropic virus HSV1 is associated with cognitive deficits in individuals with SCZ [16,17]. In addition, neuroanatomic studies have revealed reduced brain gray matter volumes among patients with SCZ exposed to HSV1 compared with healthy subjects [18,19,20]. Moreover, cognitive impairment in patients with SCZ could be associated with HSV1 infection due to the ability of HSV1 to infect neurons in brain areas involved in working memory and executive functions [21]. Thus, HSV1 could contribute to the psychotic symptoms, behavioral abnormalities, and cognitive impairment that characterize SCZ. However, given the high worldwide prevalence of HSV1 infection, it may act additively or interact with other factors, such as genetic variation [22]. For example, Prasad et al. observed that differences in gray matter variations in the prefrontal cortex among HSV1-exposed patients with SCZ and healthy controls were related to an exonic polymorphism of the MHC class I polypeptide-related sequence B gene [23]. In the present findings, multiple missense variants segregated with SCZ were associated with the HSV1 infection pathway. Taken together, we speculate that a specific genetic susceptibility to HSV1 infection may contribute to the development of SCZ. However, the associations between these gene variants and HSV1 infection are complex. Further research is needed to fully understand the relationships between these gene variants and HSV1 infection.

ECM molecules, derived from neurons and glial cells, are located in extracellular space, and brain-ECM-interaction molecules are involved in synaptogenesis and GABAergic, glutamatergic, and dopaminergic neurotransmission [24]. The brain ECM plays multiple roles in brain development, shaping synaptic plasticity, stability, cognitive flexibility, context discrimination, as well as the physiology and pathology of the central nervous system (CNS) [25]. For example, research has demonstrated the contribution of extracellular glycans and glycoconjugates, major constituents of the neural ECM, to the etiology and pathogenesis of idiopathic autism spectrum disorders and other pervasive neurodevelopmental disorders [26]. Several lines of evidence point to the involvement of the ECM in the pathophysiology of neuropsychiatric disorders such as autism, SCZ, and Alzheimer’s disease, suggesting possible etiological mechanisms linking the ECM and neuropsychiatric disorders [26,27,28]. The pathophysiology of SCZ is characterized by substantial alterations and a functional impact of the ECM’s composition and integrity [25,27,28]. Studies have demonstrated some ECM-related genes potentially associated with SCZ in animal models [6,29,30,31]. For example, a study demonstrated that patients with SCZ presented significantly lower levels of four brain ECM proteins (Lumican, a secreted acidic protein rich in cysteine, Nidogen-1, and Fibronectin) than healthy volunteers, suggesting that the brain ECM and its components are potential pharmacological targets to treat SCZ [29]. Fraser extracellular matrix complex subunit 1 (FRAS1) encodes an ECM protein that appears to regulate epidermal–basement membrane adhesion and organogenesis during development [30]. The ECM was disorganized in cortical and subcortical areas in Fras1 knockout mice, which exhibited many behavioral defects, including impaired egocentric spatial memory and aberrant olfactory learning and memory [31]. De novo mutations in an ECM gene, LAMA2, were described in patients with SCZ [6]. In addition, our previous study identified that the genetic deletion of an SCZ-susceptible gene, an activity-regulated cytoskeleton-associated protein, disturbed multiple genes involved in the ECM [32]. Here, we identified multiple missense variants within ECM-related genes transmitted by a mother with SCZ. These findings point toward associations between SCZ and multiple missense mutations within ECM-related genes, consistent with previous reports implicating a pathophysiological dysregulation of the brain ECM proteins in SCZ.

Although twin and adoption studies indicate that SZ has a genetic component with heritability [33], environmental influences are an alternative explanation for the non-hereditary portion of schizophrenia [3,34]. Epigenetic modifications, which possibly act as downstream effectors of environmental signals, not only play a regulatory role in cell differentiation and the development of the brain but also as a mechanism underlying behavioral changes [35]. Research has demonstrated that epigenetic alterations, such as altered DNA methylation, histone modifications, and RNA interference, may provide an alternative explanation for the pathogenesis of SCZ [36]. In the present study, we identified three ultrarare variants in three epigenetic-associated genes (*CYP2E1*, *NAT10*, and *CBX2*) that may be associated with the pathogenesis of SCZ. The cytochrome P450 family 2 subfamily E member 1 (*CYP2E1*) gene encodes a member of the cytochrome P450 superfamily, being considered an oxidative-stress-related gene [37]. CYP2E1 enzyme expression is differently influenced by epigenetic variation such as DNA methylation [38]. DNA methylation levels in the promoter of the *CYP2E1* gene have been associated with SCZ and tardive dyskinesia [39]. In fact, a Chinese case–control study revealed that two *CYP2E1* SNPs were associated with SCZ [40]. Thus, it should be noted that (epi)genetic variations of the *CYP2E1* gene play a role in SCZ. Further, N-acetyltransferase 10 (*NAT10*) is involved in epigenetic events, including histone acetylation and mRNA modification [41,42]. Recently, Guo and colleagues found that *NAT10* elevation in hippocampal neurons was related to anxiety- and depression-like behavior [43]. Finally, Chromobox 2 (*CBX2*) encodes a component of the polycomb multiprotein complex, which is required to maintain the transcriptionally repressive state of many genes throughout development via chromatin remodeling and histone modification [44]. Gu and colleagues reported that *CBX2* inhibits neurite development by interacting with neuro-associated genes [45]. Thus, we suggest that aberrant chromatin modulation and histone modifications due to *CYP2E1*, *NAT10*, and *CBX2* mutations may have contributed to the pathogenesis of SCZ in this family, but an additional functional study is required.

This study had the following limitations. First, we performed a whole-exome analysis in only one family. The present data may represent a specific genetic background in the sample family. Second, because we did not investigate whole-genome sequencing, several potentially valuable regions, such as non-protein-coding regions, were not sequenced.

## 5. Conclusions

We hypothesize that multiple coding variants in two KEGG pathways (HSV1 infection and the ECM–receptor interaction) indirectly affect the etiology of SCZ. With respect to rare variants, we hypothesize that damaging coding variants associated with epigenetic mechanisms are associated with SCZ, followed by ultra-rarely transmitted variants, given the evidence that newly arising mutations are enriched for more harmful mutation types. This present study provides insights with potential benefits for treating SCZ.

## Figures and Tables

**Figure 1 jpm-13-01392-f001:**
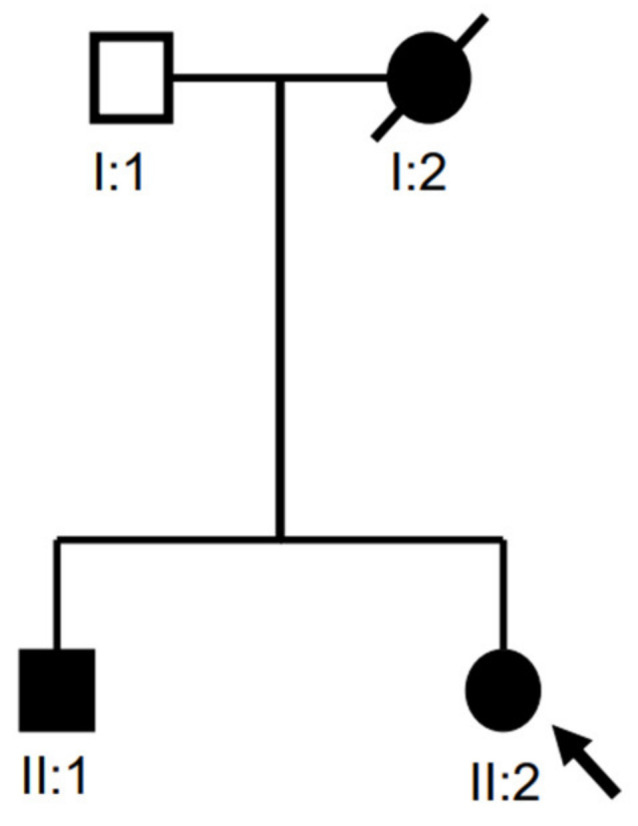
Pedigree of the sample family. Circles and squares denote females and males, respectively. Black color indicates family members with SCZ. The arrow indicates the proband.

**Figure 2 jpm-13-01392-f002:**
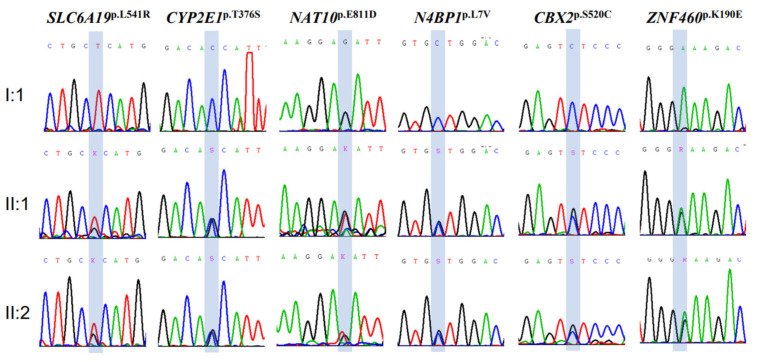
Results of Sanger sequencing of the six ultrarare variants (*SLC6A19*^p.L541R^, *CYP2E1*^p.T376S^, *NAT10*^p.E811D^, *N4BP1*^p.L7V^, *CBX2*^p.S520C^, and *ZNF460*^p.K190E^) identified in this study. The blue box indicates the position of the identified missense mutations.

**Table 1 jpm-13-01392-t001:** Protein-altering variants in the HSV1 infection pathway and ECM–receptor interactions observed in the sample family under the autosomal dominant inheritance model.

Gene	Coding: Amino Acid Change	rs Number	Public Database MAF		In Silico Analysis		
			gnomAD	Taiwan BioBank	SIFT	Polyphen-2	CADD^raw/phred^
**Herpes Simplex Virus 1 Infection Pathway ^#^**							
*ZNF573:NM_001172692*	c.A1303G:p.M435V	rs3095726	0.831	0.3895	T	B	−0.073/0.943
*ZNF177:NM_001172651*	c.C335T:p.T112M	rs2217652	0.61	0.408	T	B	−0.13/0.718
*ZNF177:NM_001172651*	c.A1363T:p.I455F	rs2230752	0.566	0.4	D	B	0.744/8.855
*ZNF132:NM_003433*	c.C755T:p.P252L	rs1465789	0.547	0.296	T	B	0.265/3.882
*ZNF132:NM_003433*	c.G608A:p.G203D	rs1122955	0.174	0.0615	T	B	−1.234/0.002
*ZNF891:NM_001277291*	c.G1375C:p.V459L	rs2173970	0.189	0.1555	T	B	1.398/15.07
*ZNF571:NM_001321272*	c.A1777G:p.K593E	rs16973890	0.0148	0.062	T	P	2.087/19.92
*SP100:NM_001080391*	c.T2477C:p.M826T	rs836237	0.797	0.4545	T	B	−1.434/0.001
*ZNF790:NM_001242800*	c.A902G:p.Q301R	rs3745775	0.273	0.245	D	B	0.255/3.759
*ZNF44:NM_001353551*	c.A394G:p.T132A	rs11879168	0.181	0.301	D	B	0.955/11.05
*ZNF529:NM_001145650*	c.T337G:p.L113V	rs2912444	0.694	0.489	T	B	−0.538/0.099
*ZNF527:NM_032453*	c.A638G:p.H213R	rs4452075	0.272	0.304	D	B	−0.35/0.245
*ZNF229:NM_00127851*	c.G1966A:p.G656R	rs1434579	0.27	0.3265	D	P	2.495/22.4
*ZNF229:NM_00127851*	c.C991T:p.R331C	rs12151338	0.274	0.3365	T	P	1.057/12.32
*ZNF568:NM_001204839*	c.G1696C:p.G566R	rs1363753	0.57	0.3325	T	B	−0.68/0.048
*ZNF302:NM_001012320*	c.A266G:p.Y89C	rs2290652	0.303	0.234	T	P	2.082/19.88
*ZNF565:NM_001366190*	c.G140T:p.G47V	rs75766177	0.00356	0.102	T	B	−0.092/0.863
*ZNF543:NM_213598*	c.C163G:p.P55A	rs6510057	0.528	0.295	T	B	0.407/5.505
*ZNF268:NM_001165887*	c.A220T:p.M74L	rs61960670	0.182	0.137	T	B	−0.68/0.049
*ZNF268:NM_001165881*	c.G2579C:p.R860T	rs62644541	0.182	0.137	D	B	2.755/22.9
*ZNF300:NM_052860*	c.5_7del:p.M2del	rs72332188	0.027317	0.038	NA	NA	N/A
*ZIM3:NM_052882*	c.A1135G:p.I379V	rs4801433	0.451	0.363	T	P	1.731/17.24
*ZIM3:NM_052882*	c.T205A:p.L69M	rs4801200	0.45	0.363	T	P	0.283/4.096
*ZIM3:NM_052882*	c.G82A:p.E28K	rs2370134	0.143	0.2665	T	P	0.174/2.83
*ZNF584:NM_001318002*	c.C298T:p.R100C	rs200779956	0.000105	0.0615	T	B	0.202/3.141
*ZNF584:NM_001363680*	c.C289T:p.P97S	rs11668789	0.178	0.061	T	B	−0.063/0.991
*ZNF484:NM_001007101*	c.G1397A:p.G466D	rs3739602	0.0239	0.2785	D	P	2.451/22.3
*ZNF440:NM_152357*	c.A371G:p.N124S	rs427880	0.515	0.3965	D	B	0.11/2.181
*ZNF440:NM_152357*	c.G1561A:p.G521R	rs117998813	0.00385	0.0755	D	B	1.105/12.82
*ZNF285:NM_001291491*	c.G158A:p.S53N	rs2571089	0.192	0.362	T	B	−0.604/ 0.071
*ZNF460:NM_001330622*	c.A568G:p.K190E	ND	ND	ND	D	P	3/23.4
*ZNF284:NM_001037813*	c.369_371del:p.S124del	rs139900131	0.473322	0.3493	NA	NA	N/A
*ZNF382:NM_001256838*	c.C1637T:p.T546M	rs61732180	0.162	0.188	T	B	0.854/9.938
*ZNF180:NM_001278508*	c.C740G:p.S247C	rs1897820	0.532	0.59	D	B	1.601/16.38
*ZNF180:NM_001278508*	c.C192G:p.C64W	rs2253563	0.517	0.5765	D	P	1.158/13.3
*ZNF180:NM_001278508*	c.T122C:p.V41A	rs2571108	0.692	0.697	T	B	0.209/3.23
*ZNF57:NM_001319083*	c.C572A:p.T191N	rs2288958	0.418	0.6085	T	B	−1.136/0.003
*ZNF816:NM_001202456*	c.C485T:p.S162L	rs11084210	0.0929	0.2365	T	B	−0.468/0.138
*ZNF30:NM_001099437*	c.A1202G:p.Y401C	rs765746	0.188	0.403	D	P	1.728/17.22
*ZNF439:NM_001348724*	c.C1013G:p.A338G	rs72994214	0.023	0.08	D	B	0.772/9.12
*ZNF317:NM_00119079*	c.G57T:p.Q19H	rs3752199	0.154	0.117	T	B	0.43/5.76
*ZNF416:NM_001353405*	c.C284T:p.S95L	rs151324898	0.000952	0.035	T	P	0.947/10.97
*ZNF735:NM_001159524*	c.C670A:p.H224N	rs4320434	0.224	0.2575	T	B	−1.599/0.001
*ZNF875:NM_001329773*	c.G1358T:p.S453I	rs3745765	0.278	0.318	T	B	−0.005/1.302
*ZNF460:NM_001330622*	c.A568G:p.K190E	ND	ND	ND	D	P	3/23.4
*ZNF333:NM_001352243*	c.C917T:p.A306V	rs3764626	0.357	0.5945	T	B	0.635/7.848
*TLR3:NM_00326*	c.C1234T:p.L412F	rs3775291	0.236	0.361	D	P	2.844/23.1
*IFNAR1:NM_000629*	c.G502C:p.V168	rs2257167	0.156081	0.354	T	P	1.153/13.25
**ECM–receptor interaction pathway ^$^**							
*LAMA2:NM_000426*	c.G1856A:p.R619H	rs3816665	0.257227	0.0725	T	B	−1.085/0.004
*LAMA2:NM_00107982*	c.C7748T:p.A2583V	rs2229848	0.370535	0.587	T	P	4.105/27.8
*VWF:NM_000552*	c.G6104A:p.G2035D	rs186806674	0.000157	0.0045	T	B	0.811/9.502
*ITGA4:NM_000885*	c.G2633A:p.R878Q	rs1143676	0.288230	0.8475	T	B	−0.899/0.014
*LAMC3:NM_006059*	c.C1564T:p.P522S	rs869457	0.282728	0.2955	T	B	0.839/9.786
*LAMC3:NM_006059*	c.A1631G:p.E544G	rs10901333	0.477113	0.471	T	B	2.173/20.7
*LAMC3:NM_006059*	c.C2308G:p.R770G	rs3739510	0.845	0.6075	T	B	−0.095/0.851
*LAMC3:NM_006059*	c.A3244G:p.S1082G	rs2275140	0.243787	0.663	T	B	0.421/5.661
*LAMA4:NM_001105206*	c.G5443A:p.V1815I	rs3734292	0.001740	0.0755	D	P	3.647/25.2
*DMP1:NM_001079911*	c.A157T:p.S53C	rs10019009	0.281	0.4205	D	P	3.056/23.5
*HMMR:NM_001142557*	c.G737A:p.R246H	rs2303078	0.037807	0.04	T	B	0.075/1.873
*FRAS1:NM_001166133*	c.T1396A:p.L466I	rs12504081	0.350818	0.3855	T	P	2.658/22.7
*FRAS1:NM_001166133*	c.A2060G:p.D687G	rs345513	0.425	0.4485	T	B	0.327/4.597
*FRAS1:NM_001166133*	c.C2450T:p.A817V	rs6835769	0.433817	0.197	T	B	0.351/4.87
*FRAS1:NM_001166133*	c.G3406A:p.E1136K	rs12512164	0.242134	0.207	T	P	1.844/18.06
*TNN:NM_022093*	c.C3467T:p.A1156V	rs2072036	0.114564	0.178	D	B	0.296/4.24
*COL6A2:NM_001849*	c.G2039A:p.R680H	rs1042917	0.478396	0.43	D	P	3.684/25.3
*COL9A1:NM_001377290*	c.A1133G:p.Q378R	rs1135056	0.382833	0.2785	T	B	0.922/10.68
*TNR:NM_003285*	c.G382T:p.A128S	rs2239819	0.241553	0.514	T	B	1.383/14.97
*COL6A6:NM_001102608*	c.G5216A:p.R1739Q	rs16830494	0.095544	0.1835	D	B	2.068/19.77

**^#^** Fold enrichment: 2.0736; Bonferroni: 0.0101; Benjamini: 0.0093; FDR: 0.0093. **^$^** Fold enrichment: 4.0982; Bonferroni: 0.0184; Benjamini: 0.0093; FDR: 0.0093; MAF: minor allele frequency; ND: not documented; NA: not applicable; T: tolerated; D: damaging; B: benign; P: probably or possibly damaging.

**Table 2 jpm-13-01392-t002:** Protein-altering variants with minor allele frequency (MAF) < 0.5% observed in the sample family under the autosomal dominant inheritance model.

Gene	Coding: Amino Acid Change	rs Number	Public Database MAF		In Silico Analysis		
			gnomAD	Taiwan BioBank	SIFT	Polyphen-2	CADD^raw/phred^
*SSX2IP:NM_001166294*	c.G1492C:p.G498R	rs191448293	0.0002900	0.001648	D	P	1.85/18.1
*SPAG17:NM_206996*	c.A4837G:p.N1613D	rs200562117	0.00008007	0.000660	T	B	0.468/6.162
*STYXL2:NM_001080426*	c.C91T:p.R31X	rs377245710	0.00005664	0.001318	NA	NA	7.113/37
*ZNF648:NM_001009992*	c.T1105C:p.C369R	rs765064028	0.00005709	0.000668	D	P	3.636/25.2
*RGSL1:NM_001137669*	c.T226C:p.F76L	rs1403931074	0.00004728	ND	D	P	3.83/26
*EMILIN1:NM_007046*	c.G2366A:p.R789Q	rs566620372	0.00007649	ND	T	P	3.552/24.9
*ARHGEF33:NM_001145451*	c.C484G:p.P162A	rs190587324	0.0001200	0.000989	T	P	2.677/22.8
*USP34:NM_014709*	c.A3444G:p.I1148M	rs769808228	0.00007653	0.002307	T	B	1.537/15.97
*CLEC4F:NM_001258027*	c.1421delA:p.K474Sfs*44	rs781870818	0.00003535	0.000330	NA	NA	NA
*STEAP3:NM_001008410*	c.G982A:p.A328T	rs376635282	0.00002651	ND	T	P	0.49/6.395
*MYO7B:NM_001080527*	c.A694C:p.I232L	rs559170047	0.0001176	0.001979	D	P	3.607/25.1
*USF3:NM_001009899*	c.A934G:p.T312A	rs192213473	0.0001033	0.000659	T	B	0.336/4.707
*FIP1L1:NM_001134938*	c.1225_1226del:p.R413Gfs*3	rs143671659	0.000160	ND	NA	NA	NA
*NDST3:NM_004784*	c.A2027T:p.E676V	rs757955201	0.00003188	0.000330	T	B	2.585/22.6
*UCP1:NM_021833*	c.G242A:p.G81E	rs555800104	0.0001272	0.000659	D	P	3.426/24.5
*SLC6A19:NM_001003841*	c.T1622G:p.L541R	rs1176698743	ND	ND	D	P	3.562/24.9
*CENPK:NM_001349368*	c.G487A:p.V163I	rs769017086	0.00008767	0.001649	T	P	2.989/23.4
*DMGDH:NM_013391*	c.T2324G:p.L775R	rs201416183	0.000003992	ND	D	P	4.262/29.2
*ZFYVE16:NM_001284237*	c.C2189T:p.T730I	rs140823634	0.00006367	ND	T	B	1.397/15.06
*SLC22A4:NM_003059*	c.442_443del:p.L148Vfs*131	rs765247850	0.00001988	0.000330	NA	NA	NA
*TCOF1:NM_000356*	c.C122T:p.A41V	rs56180593	0.002966	0.003	D	P	1.682
*CRYBG1:NM_001624*	c.G166C:p.E56Q	rs201789082	0.0005189	0.002444	D	P	3.799/25.8
*COL10A1:NM_000493*	c.T788C:p.I263T	rs200461789	0.00008876	0.000989	T	B	0.282/4.078
*CREB5:NM_001011666*	c.G671T:p.G224V	rs142716067	0.0002903	0.006338	D	P	4.015/27.1
*SEMA3E:NM_001178129*	c.G1552A:p.V518I	rs200779956	0.0001667	0.003626	T	B	1.179/13.47
*ATP6V1B2:NM_001693*	c.C7A:p.L3M	rs201057159	0.0001301	0.001252	T	P	2.71/22.8
*MTSS1:NM_001363300*	c.G1181A:p.R394Q	rs780673847	0.00001990	0.000330	D	P	3.739/25.6
*EXD3:NM_017820*	c.C707T:p.A236V	rs370748438	0.0001587	0.000660	T	B	−0.38/0.212
*KIAA1217:NM_001282769*	c.T1169G:p.V390G	rs780371689	0.00006011	0.000330	D	P	4.244/29.1
*SORBS1:NM_001290296*	c.C649G:p.R217G	rs371784791	0.0002812	0.001995	D	B	0.928/10.74
*CYP2E1:NM_000773*	c.C1127G:p.T376S	ND	ND	ND	D	B	1.605/16.41
*NAT10:NM_001144030*	c.G2433T:p.E811D	ND	ND	ND	D	P	3.284/24.1
*PACSIN3:NM_001184974*	c.C82T:p.R28W	rs185936979	0.0004014	0.002970	D	P	4.48/32
*OR8K5:NM_001004058*	c.C821A:p.A274D	rs201996541	0.00005177	0.000989	D	P	1.472/15.55
*VWF:NM_000552*	c.G6104A:p.G2035D	rs186806674	0.0004034	0.003626	T	B	0.811/9.502
*SLC39A5:NM_001135195*	c.C1517T:p.T506M	rs2293511	0.0007447	0.004293	T	B	−0.791/0.027
*DNAH3:NM_00134788*	c.G10796A:p.R3599H	rs750487156	0.0001485	0.001320	D	P	2.868/23.1
*PRSS53:NM_001039503*	c.C1004T:p.A335V	rs188342896	0.0005128	0.003955	T	B	0.433/5.786
*N4BP1:NM_153029*	c.C19G:p.L7V	ND	ND	ND	T	P	2.527/22.5
*CBX2:NM_005189*	c.C1559G:p.S520C	ND	ND	ND	D	P	4.002/27
*GDF15:NM_004864*	c.G763A:p.A255T	rs779865510	0.000008450	ND	D	P	2.955/23.3
*ZNF784:NM_203374*	c.C298A:p.P100T	rs558625495	0.0002610	0.004529	D	B	1.118/12.94
*NLRP13:NM_001321057*	c.C2764G:p.P922A	rs181274636	0.0001805	0.004285	T	B	0.173/2.827
*PEG3:NM_001369735*	c.G1736A:p.S579N	rs532173382	0.00009551	0.000989	T	P	3.266/24
*ZNF460:NM_001330622*	c.A568G:p.K190E	ND	ND	ND	D	P	3/23.4
*C20orf96:NM_080571*	c.C58G:p.Q20E	rs756843278	0.00002503	0.001978	D	P	1.474/15.57
*SALL4:NM_001318031*	c.T1262G:p.L421W	rs758373411	0.0001061	ND	D	P	3.88/26.2
*ZNF831:NM_178457*	c.G3106A:p.A1036T	rs749543585	0.0001320	0.000330	T	B	0.618/7.683
*MCM3AP:NM_003906*	c.T5555C:p.M1852T	rs182565117	0.0002757	0.004614	T	B	0.393/5.345
*IL17RA:NM_001289905*	c.G1817A:p.G606E	rs761619526	0.0001468	0.003067	T	P	−0.335/0.264
*AIFM3:NM_001018060*	c.A151C:p.T51P	rs753362705	0.00009996	0.003014	T	B	0.363/5.011
*MYO18B:NM_001318245*	c.G5399A:p.R1800Q	rs371283219	0.00004649	0.000659	D	P	4.151/28.2
*ELFN2:NM_052906*	c.G62A:p.R21H	rs368781861	0.0001428	0.002125	T	B	0.686/8.33
*SCUBE1:NM_173050*	c.G2146A:p.G716S	rs550015784	0.0001239	0.000342	T	B	−0.44/0.159
*SCUBE1:NM_173050*	c.G2141A:p.R714H	rs535650437	0.00003703	0.000341	D	P	3.831/26

MAF: minor allele frequency; ND: not documented; NA: not applicable; T: tolerated; D: damaging; B: benign; P: probably or possibly damaging.

## Data Availability

The raw data are available upon the request of the corresponding author.

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
