# Peer review of "Indicators of HSV1 Infection, ECM–Receptor Interaction, and Chromatin Modulation in a Nuclear Family with Schizophrenia"

_jpm, 2023, doi:10.3390/jpm13091392_

Round 1
Reviewer 1 Report
This manuscript, using WES, sought to investigate genetic factors contributing to the manifestation of schizophrenia in a nuclear family within which the mother and two children were affected. Overall, the paper was well-organized and the results very interesting.
General comments
-While relatively well-written, the manuscript can use some English language editing.
For example, “Sequencing was conducted on an Illumina NovaSeq 6000 platform to generate paired 150 bp reads were generated.” should be “Sequencing was conducted on an Illumina NovaSeq 6000 platform to generate paired 150 bp reads.” (lines 62, 63).
Methods
-Section 2.2. At what depth were the samples whole exome sequenced? I may not be seeing it, but this should be important to include in the main text’s methods section.
Results
-Section 3.1. I’m assuming the family was Taiwanese, but please specify the ethnic background of the family where known. This is relevant to clinicians worldwide given ethnicity-associated variability in psychiatric genetic variants.
- Section 3.1. It would be very interesting and relevant for clinicians reading your paper to know what the mother died of, and what the medications administered consisted of for the sister, and brother if he was given any (generic names, doses, duration of administration). If this makes the word count too long, this can be added to the supplementary information in the form of supplementary methods.
- Section 3.1. In addition, were there any other members of the broader family affected with schizophrenia or psychotic episodes? This would be important to know as well.
Discussion
-In order to best contextualize your HSV1 association, it would be interesting to cite a few of the papers pointing to the link between neuropsychiatric disorders and immune / inflammatory responses (including in the context of schizophrenia, but also genetically overlapping/related disorders like autism, among others). This is particularly fascinating etiologically in light of the fact that some of these disorders are triggered by a strong immune episode.
A few examples below:
https://www.ncbi.nlm.nih.gov/pmc/articles/PMC6435494/
https://www.ncbi.nlm.nih.gov/pmc/articles/PMC6952169/
https://www.ncbi.nlm.nih.gov/pmc/articles/PMC8910888/#:~:text=Neuroinflammation%20is%20associated%20with%20increased,imbalance%20occurs%20in%20this%20disorder.
In order to best contextualize your ECM proteins association, it would also be interesting to make parallels with research showing the links between other extracellular components and other psychiatric disorders.
A few examples below:
https://www.ncbi.nlm.nih.gov/pmc/articles/PMC8359380/
https://www.ncbi.nlm.nih.gov/pmc/articles/PMC5556687/
And finally, the research area linked to epigenetic modifications in psychiatric disorders is particularly fascinating and rapidly advancing. Citing relevant papers, at the end of the section discussing the CYP2E1 association, would enrich your findings.
A few examples:
https://www.degruyter.com/document/doi/10.1515/medgen-2020-2004/html?lang=en
https://www.ncbi.nlm.nih.gov/pmc/articles/PMC3181944/
Placing your work within this broader, dynamic research context would add clout to your findings and confirm putative mechanistic etiologies.
-Since you have only done a whole exome analysis, it is important to mention in the Discussion section the limitations inherent to the fact that you have not investigated the whole genome, including non-protein-coding regions. Since more and more variants are found at increasingly long distances from genic regions (100’s of kb’s), this is a critical limitation which warrants mentioning (especially since you explicitly filter out nonprotein-modifying variants).
-Finally, given your (non-Mendelian) gene analysis approach, it would be interesting to contextualize your research in the context of ongoing work on polygenic risk scores (e.g. https://www.ncbi.nlm.nih.gov/pmc/articles/PMC7396092/). This would add a certain degree of clinical applicability / translatability to your work.
See above.
Reviewer 2 Report
Dear Authors, there are several aspects that require further attention.
Minor points:
Please check again the entire manuscript and use associated abbreviations where possible (e.g. schizophrenia - SCZ, genome-wide association study - GWAS, white blood cells - WBC, grey matter volume - GMV, central nervous system - CNS) and there may be more.
Major points:
Why the family members were not diagnosed according to the DSM-5?
What are the sequences used?
What was the amplification protocol?
Nothing regarding the amplified products purification step prior to Sanger sequencing?
260/280 nucleic acid ratio following DNA extraction?
Why the mother has not been tested (she died or was dead during the study realization?)
Good luck!
The English is acceptable.
Round 2
Reviewer 2 Report
Congratulations for reviewing the manuscript according to my instructions.